# Techniques for the Investigation of Segmented Sensors Using the Two Photon Absorption-Transient Current Technique

**DOI:** 10.3390/s23020962

**Published:** 2023-01-14

**Authors:** Sebastian Pape, Esteban Currás, Marcos Fernández García, Michael Moll

**Affiliations:** 1European Organization for Nuclear Research (CERN), Esplanade des Particules 1, 1217 Meyrin, Switzerland; 2Department of Physics—AG Kröninger, TU Dortmund University, 44227 Dortmund, Germany; 3Instituto de Física de Cantabria (CSIC-UC), Avenida de los Castros, E-39005 Santander, Spain

**Keywords:** two photon absorption-transient current technique, transient current technique, solid state detectors, silicon detectors

## Abstract

The two photon absorption-transient current technique (TPA-TCT) was used to investigate a silicon strip detector with illumination from the top. Measurement and analysis techniques for the TPA-TCT of segmented devices are presented and discussed using a passive strip CMOS detector and a standard strip detector as an example. The influence of laser beam clipping and reflection is shown, and a method that allows to compensate these intensity-related effects for investigation of the electric field is introduced and successfully employed. Additionally, the mirror technique is introduced, which exploits reflection at a metallised back side to enable the measurement directly below a top metallisation while illuminating from the top.

## 1. Introduction

The two photon absorption-transient current technique (TPA-TCT) was developed in the framework of the RD50 collaboration for the characterisation of silicon detectors [1,2]. It is based on the well-established single photon absorption—transient current technique (SPA-TCT) [3,4], in which lasers with a wavelength within the linear absorption regime of silicon (<1.15 μm) and a continuous absorption along the beam propagation are used. In contrast, the TPA-TCT uses lasers with a wavelength in the quadratic absorption regime (>1.2 μm to <2.3 μm), wherefore a confined excess charge carrier density is only generated around the focal point, which enables probing of the silicon bulk with a three-dimensional resolution. Equivalent to conventional TCT methods, the drift of the generated excess charge carriers is recorded, and device quantities such as the charge collection time, the collected charge, or the electric field can be investigated; this is accomplished, however, in a three-dimensional manner. Therefore, TPA-TCT can be performed with illumination from the top, back, or edge while still maintaining the resolution along the laser beam propagation direction [5]; this is not possible for SPA-based methods. Only the so called edge-TCT, where illumination is applied from the edge, allows SPA-based methods with high enough absorption length to obtain resolution along the device depth [4]. However, illumination from the edge requires tedious alignment and usually, sample preparation methods such as polishing, which are not needed for illumination from the top and back.

Segmented devices benefit in particular from a three-dimensional resolution [6,7,8]. However, they can add complexity to the measurement because different material layers are usually involved, which can lead to reflection [9] or laser beam clipping and can thereby influence the laser intensity, i.e., the generated charge, depending on their position inside the device. Within the present study, methods for the characterisation of segmented devices are presented using a passive p-type strip CMOS detector as an example; this was developed in a collaborative project of DESY, the University of Bonn, the University of Freiburg, and the TU Dortmund University [10]. It is intended to be used as a cost-efficient device for large detector experiments due to the exploitation of the commercialised CMOS process. Further, a standard strip detector is used to investigate techniques that require illumination from the top and back side, as this device was prepared to be accessible from the top and back side.

The paper is structured in the following way. First, an overview about the used setup and the device under test (DUT) is given in Section 2; then, analysis methods that are especially useful to segmented sensors are derived in Section 3, which is followed by a characterisation of the passive strip CMOS detector with TPA-TCT and an application of the prior derived methods in Section 4. Moreover, the so-called mirror technique, which will be explained later, is applied to the standard strip detector. Finally, conclusions are given in Section 5.

## 2. Experimental Setup

A schematic of the used TPA-TCT setup is depicted in Figure 1. The setup uses the FYLA LFC1500X fibre laser module [11], which provides a 430 fs-pulse with a wavelength of 1.55 μm, a pulse frequency of 8.2 MHz, and a pulse energy of 10 nJ at its output. Downstream from the output, the light is coupled out from the fibre and traverses in open space towards the pulse management module. The pulse management module includes an acusto-optic modulator to regulate the pulse frequency and a neutral density filter to adapt the pulse energy. For the presented measurements, a pulse frequency of 200 Hz and a pulse energy of 0.22 nJ (measured below the objective) is used. It was experimentally verified that this pulse energy is below the threshold of electron-hole plasma creation, which occurs at high enough charge carrier densities [12].

Downstream from the pulse management module, the light is guided inside a Faraday cage, which is used for electromagnetic shielding. Inside the Faraday cage, the light passes a 50/50 beam splitter, where one arm reaches the DUT and the other arm reaches a silicon diode that is used as a reference to correct for potential energy fluctuation from the laser source. Highly focusing objectives are used in both arms to increase the charge generation by TPA, as it scales quadratically with the light intensity. The highest possible focusing, while avoiding aberration [5,13], is used to increase the TPA efficiency as much as possible. For the presented measurements, an objective with a numerical aperture of 0.5 is found to be ideal. The objective achieves, for the alignment at hand, a beam waist of $w_{0}=(1.63\pm 0.11)\;\mu m$ and a Rayleigh length inside silicon of $z_{R,\mathrm{Si}}=(13.07\pm 1.05)\;\mu m$ in the DUT arm. The beam parameters are measured with the knife-edge technique following the procedure described in reference [14]. The beam parameters of the reference arm were not measured as the reference’s beam parameter are not required for its operation.

The DUT is glued with silver epoxy to a passive readout board, which is mounted below the objective on a copper chuck. The chuck is thermally coupled to a Peltier element, where the hot side is cooled by a HUBER chiller. For the presented measurements, the DUT is actively temperature controlled to 20 °C, the Faraday cage is continuously flushed with dry air, and the device is illuminated from the top. The copper chuck is positioned on a six-axes HXP50-MECA stage which allows high precision movement and rotation along all three dimensional axes. The rotation is used to level the DUT and ensure an orthogonal incidence of the laser. Further information about the setup can be found in reference [14].

A passive strip CMOS detector and a standard strip detector are used to investigate the TPA-TCT in segmented detectors. The passive strip CMOS detector is a 150 μm thick p-type strip detector with a pitch of 75.5 μm and 2 cm long strips. The resistivity of the wafer is 3–5 kΩ cm and the device is fully depleted for bias voltages above 30 V. It is fabricated in a 150 nm CMOS process by LFoundry [15], with ≈1 cm${}^{2}$ big reticles that are stitched together. There are three different implantation designs available: the regular, the low dose 55 μm, and the low dose 30 μm design. All the presented measurements are performed on the low dose 55 μm design, which is shown in Figure 2. Further information about the implantation designs can be found in reference [10]. The back side of the passive strip CMOS detectors is fully metallised, wherefore reflection of the laser beam is expected.

The standard strip detector is fabricated by Micron Technologies, Inc. [16]. It is 300 μm thick, has an 80 μm pitch, 30 μm width metals, and a n-in-p implantation. The back side is fully metallised, but etching was applied to remove part of the back side metal, making the device available for illumination from the back side.

Both DUTs are biased from the p-side (rear/back side) and readout via an n-side (front/top side) alternating current (AC) pad of a strip. The AC pads of the first and second neighbouring strips, as well as the bias ring, are grounded. The signal is amplified by a CIVIDEC [17] 40 dB current amplifier and recorded by an Agilent DSO9254A oscilloscope.

## 3. Analysis Methods

In the following, the later used analysis method is derived, starting from the Shockley–Ramo theorem [18]. The Shockley–Ramo theorem describes the current induced by drifting charge carriers at the position xyz on the collecting electrode of a sensor as: (1)$$I\left(x,y,z\right)=Q\mu_{e/h}\vec{E_{w}}\left(x,y,z\right)\vec{E}\left(x,y,z\right),$$
with the generated charge *Q*, the charge carrier’s mobility $\mu_{e/h}$, the weighting field $E_{w}$ that accounts for the sensor geometry, and the electric field *E*. The charge carrier mobility itself depends on the electric field, the temperature, and the doping concentration [19]. Equation (Equation 1) is only valid for non-saturated drift velocities, and if the electric field is increased beyond the point of drift velocity saturation, the induced current will stay constant at the saturation value $I_{\mathrm{sat}}=Q\vec{E_{w}}{\vec{v}}_{\mathrm{sat}}$, with the saturated drift velocity $v_{\mathrm{sat}}$. The position dependence of Equation (Equation 1) results in a varying induced current along the excess charge carriers drift, which translates to a time-dependent induced current signal. The induced current starts with the deposition of the excess charge carriers (t=0) and ends as soon as they are collected ($t=t_{\mathrm{coll}}$). Ionising processes, like light absorption, generate excess charge carriers in pairs of electrons and holes, which have an opposite polarity and drift towards opposite collecting electrodes. Therefore, the measured induced current $I_{m}$ is a superposition of both induced currents from electrons and holes: (2)$$I_{m}\left(t\right)=I_{e}\left(t\right)+I_{h}\left(t\right).$$

Along their drift, electrons and holes experience the weighting field and the electric field corresponding to their position and therefore can induce very different currents that are superimposed. Furthermore, their distributions are smeared out by diffusion, wherefore the position information of the induced current can not be extracted without large uncertainties from the signal waveform. For this reason, Equation (Equation 1) is investigated by using the induced current right at the deposition (t=0) for various depositions at different xyz-positions. However, due to the finite response of the readout electronics, the induced current at t=0 is experimentally not accessible and, under the assumption of a linear response of the readout electronics, the induced current at a short time after the deposition $t_{\mathrm{pc}}$ is used to approximate I(x,y,z): (3)$$I_{m}\left(t_{\mathrm{pc}},x,y,z\right)\approx I\left(x,y,z\right),$$
where xyz is the position of the excess charge carrier generation. This method is known as the prompt current method [4]. With respect to the investigation of detectors and especially segmented devices, it may occur that the generated excess charge carrier distribution varies, due to, e.g., laser beam clipping and light reflection at metallisation or fluctuations of the laser source. As the prompt current method depends on the generated charge, artefacts occur if the generated charge *Q* is not the same for all positions [20]. To overcome charge dependence, we propose to weight the prompt current with the collected charge to obtain the weighted prompt current: (4)$$\frac{I_{m}\left(t_{\mathrm{pc}},x,y,z\right)}{Q_{\mathrm{coll}}\left(x,y,z\right)}\approx \frac{I(x,y,z)}{Q(x,y,z)}=\left(\mu_{e}+\mu_{h}\right)\vec{E_{w}}\left(x,y,z\right)\vec{E}\left(x,y,z\right).$$

The collected charge is calculated by the integral of the measured induced current over time: (5)$$Q_{\mathrm{coll}}\left(x,y,z\right)=\int_{0}^{t_{\mathrm{coll}}}I_{m}\left(t,x,y,z\right)dt.$$

Note that the weighted prompt current method requires that all generated charge is collected ($Q=Q_{\mathrm{coll}}$) to properly weight the prompt current. This condition is, for example, not valid if ballistic deficit is present [21] or if charge is lost or trapped by defects in irradiated devices [22]. In addition, laser beam clipping can affect the numerical aperture and thereby the focus of the laser. This effect can not be mitigated by the weighted prompt current method.

### 3.1. Extraction of the Collected Charge and Prompt Current

In the following, the extraction of the quantities from measurement data needed for the weighted prompt current method is explained. The start time of an individual waveform is extracted by linear regression of the signal’s rising edge. In a second iteration, the mean of all the individual start times of waveforms with a high signal-to-noise ratio (SNR) is taken as the start time t=0 that is used to extract the generated charge of the induced current after Equation (Equation 5). Taking the mean of the start times from waveforms with a high SNR improves the stability of the method because the fitting procedure can fail if the waveform’s SNR is not high enough. A collection time of $t_{\mathrm{coll}}=10\;ns$ is used, as it is found suitable for the DUT to collect all the generated excess charge under the used measurement conditions, i.e., bias voltage and temperature. Note that $t_{\mathrm{coll}}$ strongly depends on the device and the measurement conditions. The choice of the prompt current time $t_{\mathrm{pc}}$ is a compromise between the justification of Equation (Equation 3) and the SNR. Shorter $t_{\mathrm{pc}}$ improves the justification of the approximation, but decreases the SNR. Following reference [4], the current after $t_{\mathrm{pc}}=600\;ps$ is used as the prompt current, which is found to be a satisfactory compromise. Figure 3a shows an example waveform with the extraction of the prompt current and the collected charge.

### 3.2. Extraction of the Depletion Voltage and Device Thickness

Beyond investigation of the electric field, the TPA-TCT can be used to measure the depletion voltage and device thickness. Those parameters are usually extracted from a scan along the device depth for different bias voltages. If no laser intensity-varying effects are present, the resulting charge profile $Q_{\mathrm{coll}}\left(z\right)$ can be fitted with: (6)$$Q_{\mathrm{coll}}\left(z\right)=C\left[\arctan\left(\frac{d-(z-z_{\mathrm{off}})}{z_{0}}\right)+\arctan\left(\frac{\left(z-z_{\mathrm{off}}\right)}{z_{0}}\right)\right],$$
with the depletion depth of the device *d*, the offset from the *z*-axis origin $z_{\mathrm{off}}$, the Rayleigh length $z_{0}$, and a constant *C* that summarises material properties, setup properties, and various other constant factors [14]. The depletion depth is directly provided by Equation (Equation 6), and the depletion voltage is found as the voltage for which the extracted depletion depth is not increasing anymore. The device thickness is the value that the depletion depth asymptotically approaches for increasing bias voltages. The procedure for the depletion voltage is sensitive to the used collection time $t_{\mathrm{coll}}$, as the device can collect all the charge by diffusion; if it is close to full depletion and if the collection time is long enough. Therefore, a careful analysis of the collection time is needed to properly extract the depletion voltage with this method. When intensity-varying effects are present, the fitting of Equation (Equation 6) is usually unfeasible along the whole device depth, and the method can not be applied. Alternatively, the depletion voltage can be extracted from the time over threshold profile, which shows the time that the induced current signal is above a given threshold. The extraction method is shown in Figure 3b for a threshold of 15% of the signal’s maximum. The threshold is calculated for each waveform individually in order to avoid the influence of varying laser intensity, because a signal with a given shape will cross a fixed threshold faster than a signal with the same shape but a smaller amplitude [23]. This effect is known as the time-walk effect. In under-depleted bulk, excess carriers induce current due to movement by diffusion, which is non-directional and therefore much slower than drift in depleted bulk. This leads to a less steep and therefore longer signal [24], which results in a steep rise in the time over threshold profile. The depletion voltage can be found as the voltage at which no influence of diffusion is present in the time over threshold profile. When the DUT provides reflection at the back side, the time over threshold profile can be used to extract the position of the DUT’s rear side, as the profile is mirrored by this reflection [9]. The closer the excess charge carriers are generated to the back side, the further the electrons need to drift towards their collection n-electrode at the top side. Thus, the time over threshold increases until the reflection mirrors the profile; therefore, the position of the back side aligns with a peak in the time over threshold profile. The position of the front side surface is conveniently extracted from the collected charge profile by fitting Equation (Equation 6) to only the first part of Q(z). The device thickness is the difference between the back side and the top side position.

## 4. Results

In the following, the results of a yz-scan across the readout strip of the passive CMOS strip detector are analysed and discussed. The *y*-axis is oriented perpendicular to the strip metal orientation, and the *z*-axis points outside the active volume. Figure 4 shows the collected charge for the yz-scan for a bias voltage of 100 V. It can be seen that the profile is structured and non-homogeneous below the readout strip. The two triangular shaped structures originate from clipping at the top side metallisations of the p-stops and the strip, and the plateau of charge collection at $z_{\mathrm{Si}}=-0.31\;mm$ occurs due to reflection at the back side metallisation. This is not a feature of the device, but solely an artefact of measurement that complicates the interpretation of the data.

To extract the depletion voltage and the device positions, scans along the device depth were performed for different bias voltages at a stage *y* of 0.07 mm (compare to Figure 4). The time over threshold profiles of these scans along the device depth are shown in Figure 5a. The time over threshold for 20 V increases steeply at $z_{\mathrm{Si}}<-0.25\;mm$, which indicates the presence of diffusion and, thus, an incomplete depletion of the device. The device is fully depleted for all the other bias voltages, as no signs of diffusion are found for bias voltages ≥ 30 V. The position of the back side surface is extracted from the peak in the rear of the time over threshold profiles, which is found at $z_{\mathrm{back}}=-0.31\;mm$. The position of the top surface is determined in Figure 5b from a fit towards the top side of a charge collection profile taken after full depletion. Only the top side part is fitted, because the charge collection profile is affected by clipping at the top surfaces and reflection at the back surface, which is later discussed in more detail. The measured Rayleigh length $z_{0}=13.2\;\mu m$ is within the errors in agreement with the Rayleigh length measured with the knife-edge technique. The position of the top side is determined as $z_{\mathrm{top}}=z_{\mathrm{off}}=-0.154\;mm$. Therefore, a total device thickness of d=156 μm is found, which is in agreement with the expected thickness.

Figure 6 shows a comparison between the prompt current (Figure 6a) and the weighted prompt current (Figure 6b) for the same yz-scan of Figure 4. As expected, the prompt current profile contains similar structures to the collected charge profile, because the prompt current is sensitive to changes in the collected charge. These artefacts are mitigated by the weighted prompt current because it compensates for variations in the collected charge and, therefore is not impacted by laser clipping at the top side metallisations, reflections at the back side metallisation, or any other intensity-varying effect that might occur. The analysis is performed according to Equation (Equation 4), and the time $t_{\mathrm{pc}}=600\;ps$ is used. The observed artefacts of the charge profile fully disappear, and the volume below the readout strip is much more homogeneous. Furthermore, it can be seen that the reflection contains meaningful information about the detector as it is the mirror image of the DUT region. The reflection contains a clean picture even below the top side metallisation, wherefore this technique can be used to investigate the electric field directly below the metal with top illumination. This method will be referred to as the mirror technique, and the idea behind it is schematically shown in Figure 7.

The mirror technique is only available in TPA-TCT and not in SPA-TCT, because three-dimensional resolution is required. Note that the laser beam is always clipped by the top side metal and the clipping will vary with the position. Therefore, only light intensity-independent quantities, e.g., the collection time, the weighted prompt current, or the rise time, are available to be investigated with the mirror technique. This technique is, depending on the metallisations, available for illumination from the top or back, but not from the edge. Furthermore, the numerical aperture can be affected by the clipping, wherefore the focus of the laser can vary from the focus in the ordinary image. To further investigate the mirror technique, a standard strip detector is used, as illumination can be employed from the back side to directly illuminate below the strip metal, and illumination can be employed from the top side. For the top side illumination, the reflection at the back side metal can be used to create the mirror image, as previously shown with the passive strip CMOS detector in Figure 6b. The device is fully depleted for bias voltages > 35 V and the presented measurements are performed at a bias voltage of 50 V.

Figure 8 shows the weighted prompt current of yz-scans across the readout strip measured with illumination from the back side ( Figure 8a), and the mirror image that is obtained with illumination from the top side ( Figure 8b). It can be seen that the images agree qualitatively, as the maximum weighted prompt current is found in the same position for both images. However, Figure 8b appears to have less resolution along the *z*-axis and shows the features in different dimensions. This is a direct effect of the clipping, as the clipping lowers the numerical aperture, which leads to an increased Rayleigh length and, therefore, less resolution along the *z*-axis. The same scaling factor of 3.77 to correct for the refraction of silicon is used for both images, which leads to an overestimation of the *z*-dimension in the mirror image because the decreased numerical aperture also leads to a decreased scaling factor. However, the image still contains the same information as the non-mirror image in Figure 8a, although that image was obtained with back side illumination, which is usually not feasible as the back side of such devices is typically metallised.

## 5. Conclusions

In this paper, the weighted prompt current method was introduced and employed on a passive strip CMOS detector. It mitigates the intensity dependence of the prompt current method, which is useful for the investigation of segmented devices, because reflection, laser beam clipping, or any other laser intensity varying effect can be mitigated by this method. In a yz-scan of a passive strip CMOS detector, artefacts from clipping and reflection were observed in the collected charge profile. These artefacts fully disappear in the weighted prompt current profile, which enables investigation of the electric field with the TPA-TCT in such devices. Furthermore, the mirror technique was introduced and employed in a standard strip detector. This technique exploits the reflection at the back side metallisation to probe directly below the top side metals. Additionally, due to position-dependent light clipping, only intensity-independent quantities can be investigated with this method. The above introduced techniques are not only suitable for the investigation of strip detectors, but can be employed on any segmented or unsegmented detector.

## Figures and Tables

**Figure 1 sensors-23-00962-f001:**
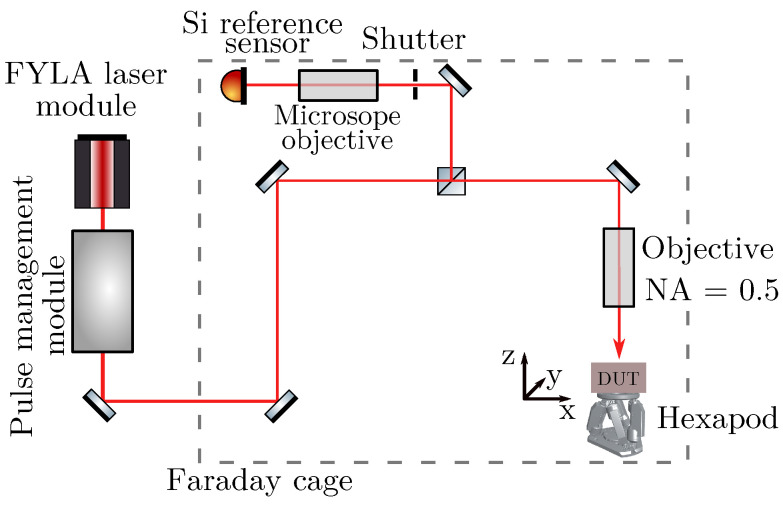
Schematic drawing of the used TPA-TCT setup.

**Figure 2 sensors-23-00962-f002:**
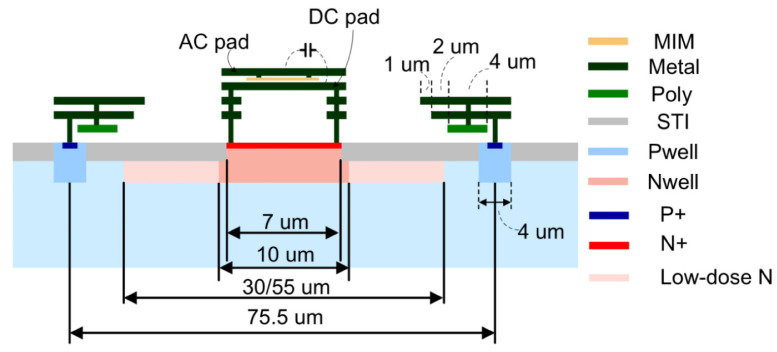
Schematic drawing of the low dose design implantation of the passive strip CMOS detector. MIM and STI stand for metal-insulator-metal and shallow trench isolation, respectively. Reprinted with permission from Ref. [10]. 2022, Elsevier.

**Figure 3 sensors-23-00962-f003:**
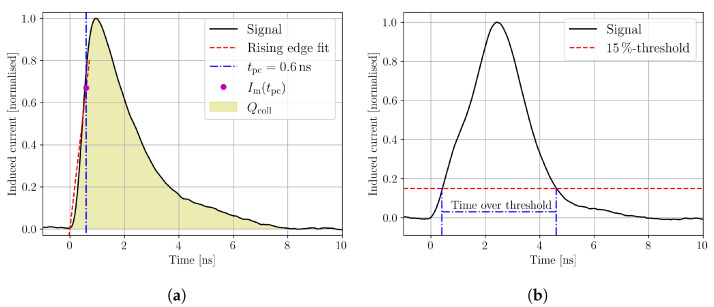
Examples of the induced current signal with corresponding analysis to extract the start time, the prompt current, and the collected charge (**a**) and the time over threshold (**b**).

**Figure 4 sensors-23-00962-f004:**
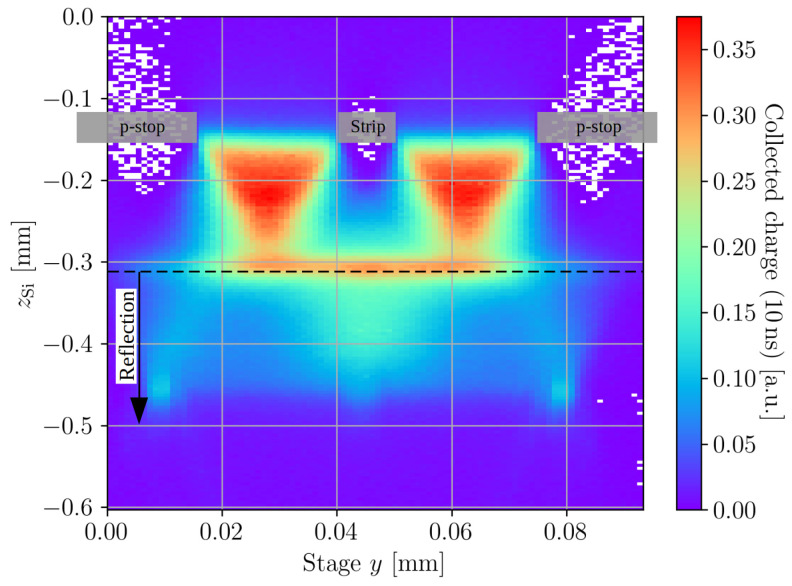
Charge collected within 10 ns for a yz-scan, taken at a bias voltage of 100 V. The scan is performed between the middle of the p-stop left of the readout strip (y=0 mm) and the middle of the right p-stop (y=0.9 mm). The top side metallisation is located at $z_{\mathrm{top}}=-0.154\;mm$ and the backside metallisation is indicated by the black dashed line at about $z_{\mathrm{back}}=-0.31\;mm$. All the signals acquired behind the back side metallisation are attributed to a focused reflection at the back side metal. The readout strip’s metal and the metals above the p-stop are shown schematically by the transparent grey boxes.

**Figure 5 sensors-23-00962-f005:**
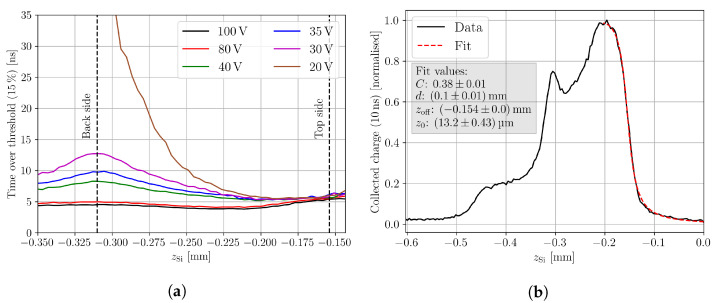
Figure (**a**) shows the time over threshold against the device depth for different bias voltages and a stage y=0.07 mm (compare to Figure 4). A total of 15% of the induced current signal’s maximum is used as a threshold. The top and back surface position are indicated by the dashed lines. The position of the back surface is extracted from the time over threshold profiles, and the position of the top surface is extracted from a fit towards the charge collection profile shown in Figure (**b**). The scan is performed at the same stage y=0.07 mm, and a bias voltage of 100 V is used. The fit is performed according to Equation (Equation 6) towards the top side. Note that the depletion depth *d* is not properly fitted because only the rising edge is fitted. Due to clipping, the rest of the charge profile is not used for the fit.

**Figure 6 sensors-23-00962-f006:**
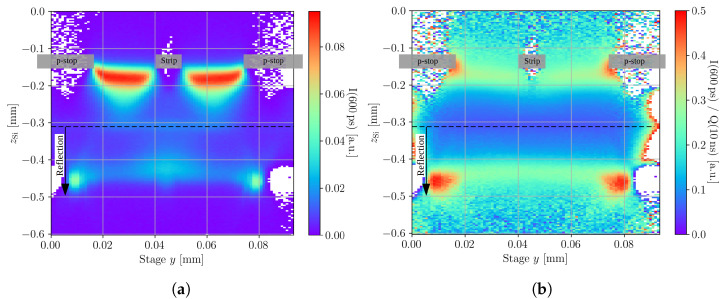
The prompt current (**a**) and the weighted prompt current (**b**) of the same yz-scan as in Figure 4 are presented. In contrast to the prompt current and the collected charge profile, the weighted prompt current profile does not show intensity related artefacts. The devices back side is indicated by the black dashed line and the readout strip’s metal and the metals above the p-stop are shown schematically by the transparent grey boxes.

**Figure 7 sensors-23-00962-f007:**
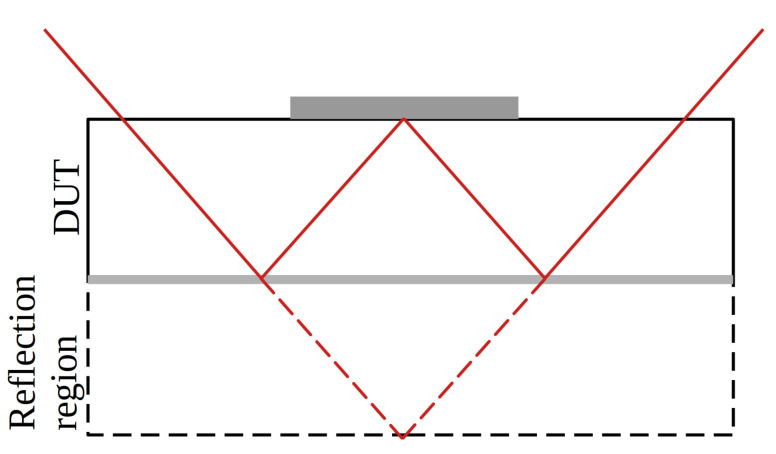
Schematic of the mirror technique using orthogonal laser beam incidence. The reflection enables probing below the strip metallisation with illumination from the top. The path of the outermost light rays of the laser are shown as a continuous red line. The dashed red lines show the continuation of the light rays beyond the metallised back side. Metallisations are depicted in grey.

**Figure 8 sensors-23-00962-f008:**
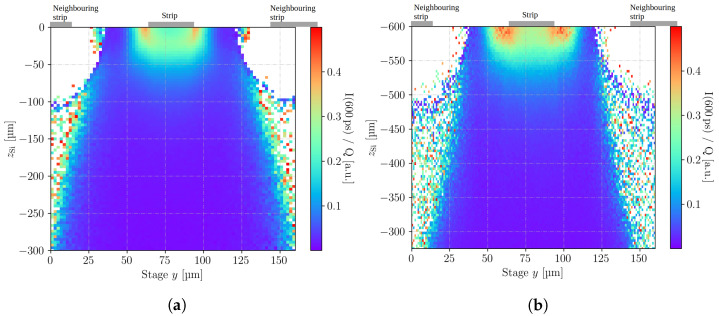
Weighted prompt current of yz-scans across the readout strip of the standard strip detector at 50 V, obtained with illumination from the back side (**a**) and the top side (**b**). Figure (**b**) only shows the mirror image and not the non-reflected image, to ease the comparison with figure (**a**). Note that the $z_{\mathrm{Si}}$-axis of figure (**b**) is slightly larger compared to figure (**a**) to compensate for the decreasing numerical aperture due to clipping. The strip metallisations are indicated by the grey boxes and note the different scale of $z_{\mathrm{Si}}$.

## Data Availability

The data presented in this study are available on request from the corresponding author. The data are not publicly available due to the complexity of the data and to maintain an overview on usage.

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
