# Peer review of "Techniques for the Investigation of Segmented Sensors Using the Two Photon Absorption-Transient Current Technique"

_sensors, 2023, doi:10.3390/s23020962_

Round 1
Reviewer 1 Report
Dear all,
amazing and very interesting work. My most sincere congratulations.
I have really very small comments/suggestions for you:
1) line 40: " ... Device Under Test (DUT) ... "
2) line 68: You mentioned that the beam parameters of the reference arm were not measured. Is there a specific reason for?
3) line 79: " ... detector ... "
4) line 82 and 89: i suggest to make a reference or specify somewhere the involved foundries LFoundry and Micron (I suppose this last one is Micron Technologies, Inc?)
5) from line 97: Somehow your .pdf lost the line indication for half a page
6) equation (1): Considering that the S.R. theorem has a dot product in it, I suggest to move the mobility at the left side of the weighting field vector (QμE*v)
7) lines 98-99: How do you check that all charge is collected?
8) line 114: "Shorter t_pc improves ..." (maybe my error, sorry in advance if yes)
9) line 135: "In under depleted ... "
10) line 200: "... CMOS detector ... "
11) figure 2 and 4: maybe figure a bit larger in my opinion but not necessary

Reviewer 2 Report
The article describes the application of a TPA-TCT techniques to a CMOS sensor and to a standard Si-strip device.
The technique allows to perform 3-D studies of several properties of the device, and will proof of great value in the
investigation/development of silicon sensors.
I congratulate for the quality of the work and the results achieved.
The article is, in my opinion well written and generally clear. The setup and the analysis
techniques are well presented and the results significant. The quality of the plots and pictures is suitable for publication.
The article fully match the scope of the journal.
Please find below a list of comment/suggestions, to be implemented in a minor revision of the manuscript
Line 20: "...the charge carrier drift of the generated excess charge carriers...". I suggest to rephrase as "...the drift of the generated excess charge carriers..."
Line 24: "... laser beam propagation.. " -> "... laser beam propagation direction... "
Line 23-25: reading the sentence looks like the SPA have spatial resolution along the laser propagation direction if
the illumination is done from the side, which means in the XY plane, not in the z plane. Please adjust.
Line 45: "...results are concluded..." -> I suggest "...conclusions are given..."
Section 2: Several times the word "behind" is used in the description of the setup, which is to me a bit confusing.
Please consider using "downstream" or equivalent.
Line 79: "detecor" -> "detecTor"
Line 83: "It has been...performance". This sentence has no use in the paper and distract the reader. Should be removed.
I think only one of the two reference should be retained, being used later.
Line 87. Please consider removing the word "moreover"
Line 96: Reference 16, "cividec" -> "Cividec" or "CIVIDEC"
Section 3: Line numbers disappeared...
Section 3: up to equation 4 it is (should be) all well known material. Try to summarize more. For example you can provide directly eq.3
and just mention the saturation velocity.
Line ~98: reference 17-18: One reference to the Ramo thorem is enough. Please remove one of the two.
Line 107-109: I don' understand why it is needed to compute the average of the t0. Is it not enough to use the individual t0 as starting point for the the current and charge measurements?
Lines 122-124: The sentence is a bit confusing, as the device thickness is fix and cannot change. Replacing "..device thickness..." with "...depletion depth.." or similar and stating that "the asymptotic value is the device thickness" would make the sentence, in my opinion, much more clear.
I suggest the same change in line 120 ("thickness of the device" -> "depletion depth of the device").
Figure 7: looking at the figure looks like a inclined illumination, paired with the mirror techniques, could avoid the clipping effect of the strip metal. In the article looks like you always used an orthgonal beam. Did you try or plan to try at different angles?
It would be good to add a sentence on this.
Figure 8b: the z scale is a bit confusing. The picture is mirrored for better comparison, but I suggests to mirror also the scale (-600 on top, -300 on bottom)
